# The Oral Cavity—Another Reservoir of Antimicrobial-Resistant *Staphylococcus aureus*?

**DOI:** 10.3390/antibiotics13070649

**Published:** 2024-07-14

**Authors:** Marek Chmielewski, Oliwia Załachowska, Dominika Komandera, Adrian Albert, Maria Wierzbowska, Ewa Kwapisz, Marta Katkowska, Alina Gębska, Katarzyna Garbacz

**Affiliations:** 1Oral Microbiology Student Scientific Club, Medical Faculty, Medical University of Gdansk, 80-204 Gdansk, Poland; m.chmielewski@gumed.edu.pl (M.C.); o.zalachowska@gmail.com (O.Z.); d.komandera@gumed.edu.pl (D.K.); texen96@gmail.com (A.A.); 2Department of Oral Microbiology, Medical Faculty, Medical University of Gdansk, 80-204 Gdansk, Poland; maria.wierzbowska@gumed.edu.pl (M.W.); ewa.kwapisz@gumed.edu.pl (E.K.); marta.katkowska@gumed.edu.pl (M.K.)

**Keywords:** *Staphylococcus aureus*, MRSA, oral carriage, dentistry students, antimicrobial resistant

## Abstract

*Staphylococcus aureus* is one of the most common potentially pathogenic bacteria that may asymptomatically colonize many sites of healthy carriers. Non-nasal carriage, especially in the oral cavity, and its role in transmitting antimicrobial-resistant *S. aureus* strains in the healthcare community, is poorly understood. This study aimed to assess the prevalence and antimicrobial susceptibility of *S. aureus* in both oral and nasal cavities among preclinical dentistry students. A total of 264 oral and nasal swabs were taken from 132 participants, and all specimens were cultured using standard diagnostic procedures and antimicrobial susceptibility testing (EUCAST). The prevalence of *S. aureus* exclusively in the nasal (11.4%) or oral (9.1%) cavity was comparable, while concurrent oral and nasal carriage was present in 27.3% of participants. Although antibiotic resistance rates observed in both oral and nasal isolates were similar (ranging from 2.7% to 95.5%), 16.7% of carriers exhibited distinct antibiotic resistance profiles between oral and nasal isolates. Three (2.7%) methicillin-resistant *S. aureus* (MRSA) were isolated from the mouth and nose but multidrug resistance (27.3%) was more frequent in the oral than in the nasal isolates: 34% and 21.1%, respectively. This study demonstrated that preclinical dentistry students have a similar rate of oral *S. aureus* carriage as the nasal carriage rate, and that the oral cavity can be colonized by antimicrobial-resistant strains that do not originate from the nose. Consequently, the oral cavity seems to be an unjustly overlooked body site in screening for *S. aureus* carriage.

## 1. Introduction

*Staphylococcus aureus* is one of the most common pathogens, and it frequently and asymptomatically colonizes healthy carriers (about 30%) [1]. It is responsible for various infections, from mild soft skin infections to severe ones like endocarditis and sepsis, or even fatal incidents [2]. Staphylococcal infections may be awkward to treat and carry a risk of failure, especially those caused by methicillin-resistant *S. aureus* (MRSA). MRSA causes healthcare-associated (HA-MRSA) and community-associated (CA-MRSA) infections. In the last two decades, a growing number of infections caused by non-nosocomial MRSA, community-acquired MRSA (CA-MRS), was observed. CA-MRSA genetically differs from HA-MRSA, is less resistant to non-β-lactam antibiotics, and carries a smaller version of staphylococcal cassette chromosome mec (SCC*mec*) [3]. 

Although anterior nares are considered a primary host site of *S. aureus*, the oral cavity can be colonized by *S. aureus* with a high frequency of up to 84%. The oral cavity is colonized by numerous and highly diversified microbiota. The properties of the mouth, as a microbial habitat, are dynamic and dictate the types of bacteria able to persist, so only some of them can colonize the oral cavity [4]. *S. aureus* was found on the oral mucosa, dorsal surface of the tongue, dental plaque, saliva, denture surface, angular cheilitis, and even periodontal pockets [5,6,7,8,9,10,11,12,13]. Although the presence of *S. aureus* in the oral microbiota seems to be a consequence of migration from the nasal cavity, this is unclear [14]. Some reports demonstrated the oral cavity as an independent reservoir of *S. aureus*, with a crucial role in disseminating to distant body sites, such as the bloodstream or the lungs [2]. 

Healthcare workers, including students, are *S. aureus* carriers. The percentage varies with the group studied, and the ratio can be more than 50% [15,16]. In dentistry, there is a higher risk of cross-transmission of many bacteria, including *S. aureus,* between the patient and the dentistry personnel. The most common transmission route is indirect via droplets by inhalation of microbes released to the surroundings from dental units, rarely by direct contact [17]. Despite numerous studies describing *S. aureus* carriage among healthcare workers, dentistry students must be sufficiently tested, and data are scarce. There are only recommendations for examination of anterior nares as a primary site of *S. aureus,* including MRSA, so most of the studies mainly focused on them [18,19,20,21,22,23,24,25,26,27,28]. Other body sites like the oral cavity, nasopharynx, or hands were chosen rarely [6,8,29,30,31,32]. Besides the human body, some authors tested the medical or dentistry environment as a place where *S. aureus* can survive [29,30,33,34]. 

Limited data exist about nasal and oral carriage in the same person among dentistry students. This study assessed the prevalence and antimicrobial susceptibility of nasal and oral *S. aureus* carriage among preclinical dentistry students before exposure to the healthcare environment and clinical experiences.

## 2. Results

The studied population included 132 preclinical dentistry students, from 1st year—67 students (50.8%) and from 2nd year—65 students (49.2%). The participants were mainly female, 78% (n = 103, 55 were from the first year and 48 from the second); male (n = 29, 1st—12, 2nd—17). In total, 27.7% of the students exhibited the simultaneous nasal and oral carriage of *S. aureus*, and significantly less nasal only (11.4%) or oral only (9.1%). The percentage of first- and second-year carriers was similar, 29.9% and 24.6%, respectively (Table 1). Interestingly, although more female students participated in the study, the male students (69%) statistically were more often colonized than females (41.7%) (*p* < 0.05). 

In total, 110 *S. aureus* were isolated from 63 (47.7%) carriers, either from the nose, from the oral cavity, or both cavities simultaneously. *S. aureus* isolates were slightly more frequently found in the nasal habitat than in the oral, 57 to 53 (51.8% vs. 48.2%). 

Similarly, non-significant differences were found in the number of *S. aureus* isolates, 58 and 52, respectively, among carriers from first- and second-year students. In the first group, the ratio of nasal to oral isolates was 27.3% to 25.5%, while in the second group, it was slightly lower at 24.5 vs. 22.7% (*p* > 0.05).

Regarding antibiotic resistance, *S. aureus* isolates were resistant to penicillin (95.5%), tetracycline (47.3%), erythromycin (28.2%), clindamycin (19.1%), gentamycin (15.5%), and trimethoprim-sulfamethoxazole (2.7%). All isolates were fully susceptible to ciprofloxacin and chloramphenicol (Table 2). Only three (2.7%) methicillin-resistant *S. aureus* (MRSA) isolates were found in two carriers (3.2%). One participant carried MRSA in both the nose and mouth, and the second was colonized only in the nose. All MRSA isolates were resistant to penicillin, tetracycline, erythromycin, and clindamycin. Besides methicillin resistance, phenotype MLS_B_ (macrolide-lincosamide-streptogramin B) resistance was detected in 16.4% of oral and nasal isolates, 10% and 6.4% respectively. Considering the origin, the nasal and oral strains showed very similar antibiotic resistance rates, ranging from 2.7% to 95.5%; the differences were not significant (*p* > 0.05). The remaining antibiotic-resistance rates are summarized in Table 2.

Multidrug resistance (MDR) exhibited 30 (27.3%) isolates, all MRSA (3), and 27 (25.2%) MSSA. MDR isolates were detected more frequently in the mouth (34%) than in the nose (21.1%). Multiplicity (10) of antibiotic resistance (AMR) profiles of *S. aureus* isolates were found, and P-TE was the most common. Regarding the origin, only two profiles were unique; one profile was only found in the nasal cavity (E-CN-P), and the second one in the oral cavity (E-DA-CN-P-TE). Both AMR profiles were MDR. Although the majority of oral and nasal profiles in simultaneous carriers were similar, 6 among 36 (16.7%) carriers exhibited distinct antibiotic resistance profiles between oral and nasal isolates that may indicate their disparate sources.

## 3. Discussion

The study focused on an estimate of nasal and non-nasal carriage of *S. aureus* among preclinical dentistry students. Our previous screening study demonstrated that 21% of medical students at our university were *S. aureus* carriers [18]. The current study showed a higher carriage rate (47.7%). Thus, the frequency of *S. aureus* colonization has increased over the past ten years, which may depend on many factors. The prevalence of *S. aureus* depends on the studied population, country, site, and sampling methodology. Unlike our results, data from Turkey, Malaysia, and Italy demonstrated much lower carriage rates of 3.9%, 17.3%, and 20.9%, respectively [8,16,35]. In other studies from Iran (44.6%) and Sweden (50%), the results were consistent with ours [15,36].

Among the studied students, 11.4% were exclusive nasal carriers and 9.1% were oral. This result is in line with the findings from Italy: 11.9% in the nostrils and 10.4% in the mouth [8]. Similar rates were also noted in the population in Portugal, in the nasal (13.9%) and oral (12.0%) habitats [11]. However, the value for a Swedish group of oral carriers was much higher (44.6%) than ours [36]. It should be stressed that as many as 27.7% of our students had *S. aureus* in both tested sites, and the carriage in the oral cavity was very similar to that in the nose (51.8% vs. 48.2%). This is an interesting result, as there were no data about these carriers in dentistry. In the literature, the nasal cavity is still the most frequently studied for *S. aureus* colonization [19,23,24,25].

Regarding participants’ sex, our data suggest a higher incidence in males. The finding could have been more consistent, depending on the study. On the one hand, several authors, Wong, Efa, and Martinez-Ruiz, found no statistical significance in carriage between male and female groups [16,22,31,37]. On the other hand, our results were consistent with Cavaco-Silva et al. [38]. 

The methicillin resistance level of *S. aureus* in invasive infections in Poland is about 14–16%, comparable to the population-weighted European average of 15.8% [39]. The rates of MRSA carriage in students vary widely by country and year of study (0–51.4%). Studies demonstrated that clinical students are more likely to be carriers due to prolonged exposure and patient interactions [15,16,19]. Among our preclinical dental students, 3.2% were colonized with MRSA strains detected at both sites, nasal and oral. Compared to others, our values were lower than those in an Italian study (3.2%—nasal and 4.5%—oral) or an American study (6.6%) [8,30]. On the other hand, the overall estimated percentage (1.5%) in a multicenter study from Europe (Sweden, Italy, Greece, and the Netherlands) was lower than in our results [32]. However, comparing the percentages from each country, the data from Italy showed a high carriage of MRSA (7.46%); it was more often detected in the mouth and on the hands than in the nose. In contrast, among carriers from Greece or the Netherlands, the percentage of MRSA was lower, at 2.04% and 0.56%, respectively, with MRSA most often colonizing the nostrils [32]. Considering that our students had no contact with the clinic and patients, the revealed MRSA carriage indicates the need for awareness of the risk of spreading antimicrobial-resistant *S. aureus*, especially MRSA strains.

This study’s general antimicrobial resistance rates ranged from 2.7% for trimethoprim-sulfamethoxazole to 95.5% for penicillin. MSSA isolates showed the highest resistance rates towards penicillin (95.3%) and tetracycline (45.8%), while all MRSA isolates were resistant to several antimicrobials. This trend was observed in many studies [10,22,32,40]. Notably, almost 30% of our isolates presented multidrug resistance, with predominant isolates from the oral cavity (34%). Arredondo et al. reported the high prevalence of multi-resistant species isolated from the oral microbiota [41]. Our study revealed the dominance of one multi-resistant profile in both nasal and oral cavities (E-DA-P-TE) with an exclusive occurrence of single profiles at one of the colonization sites. The E-CN-P profile was solely found in the nostrils, while the E-DA-CN-P-TE profile colonized only the oral cavity. It should be stressed that 16.7% of concurrent carriers showed different antibiotic resistance patterns between oral and nasal isolates. Capos et al. also reported concurrent carriers, at a percentage higher than ours, as high as 60% [11]. It suggests that *S. aureus* isolates colonizing the oral cavity do not necessarily originate from the nostrils and that the oral cavity may be an independent reservoir of antimicrobial-resistant isolates of *S. aureus,* which needed to be confirmed by *spa* typing. 

The other authors also described the high resistance rates to penicillin and tetracycline observed in our study [10,20,22,42]. In previous studies, Efa et al. and Sharma et al. found a higher tetracycline resistance rate of 54.9% and 77.7%, respectively [22,43]. To date, tetracycline resistance has been reported mainly in strains isolated from animal sources rather than from the oral cavity [44]. Tetracycline resistance is a marker of livestock-associated *S. aureus* infections and is caused by its widespread use in antibiotic prophylaxis and animal husbandry [44]. Tetracycline is also used in dentistry to treat dental infections, especially in patients with periodontitis. Accordingly, Kwapisz et al. observed high tetracycline resistance of *S. aureus* isolated from dental patients [10]. However, Arredondo et al. recently reported the spread of this resistance in various oral bacterial species from healthy individuals [41]. Thus, oral microbiota, including *S. aureus* strains, can easily acquire resistance to tetracycline. The emergence of oral tetracycline-resistant strains seems unclear, and further studies are needed.

It is important to note that our study has limitations. The use of swabs and epidemiologic data from a singular region of Poland, as well as the absence of a molecular typing method, may impact the generalizability of our findings. Additionally, the persistent and intermittent carriers were not distinguished. However, we believe these limitations do not detract from the value of our research.

## 4. Materials and Methods

### 4.1. Study Population

The study included 132 healthy preclinical dentistry students (1st and 2nd year of study—without clinical experience yet) attending the Medical University of Gdansk (MUG) in Poland, aged 20–30. Students who agreed to participate in this study provided written consent. There were 67 (50.8%) and 65 (49.2%) participants from 1st and 2nd year students, respectively. The study was approved by the Independent Bioethics Committee for Scientific Research at the Medical University of Gdansk (no. NKBBN/500/2020).

### 4.2. Isolation of Staphylococcus aureus Strains

A total of 264 swabs from the nostrils and oral mucosa were collected from 132 dentistry students. Informed consent was obtained from all the subjects. A sterile cotton swab was used to take saliva specimens from the oral cavity and the nostrils. All the swab samples were plated onto Columbia blood agar (GrasoBiotech, Starogard Gd., Poland) and mannitol salt agar (bioMérieux, Marcy l’Etoile, France) and incubated at 37 °C 24–48 h aerobically. Suspected staphylococcal colonies were identified by standard methods based on colony characteristics, pigment production, Gram-staining, hemolysis, and a Pastorex StaphPlus latex agglutination kit (Bio-Rad, Marnes la Coquette, France). After final identification, the isolates were stored at −80 °C in Trypticase Soy Broth (Becton Dickinson, Franklin Lakes, NJ, USA) supplemented with 20% glycerol.

### 4.3. Antimicrobial Susceptibility

The antibiotic susceptibility of isolated *S. aureus* was performed on Mueller–Hinton agar (BTL, Warszawa, Poland) by the disc diffusion method and interpreted according to EUCAST [45]. The following antibiotics were tested: cefoxitin (30 μg), clindamycin (2 μg), ciprofloxacin (5 μg), chloramphenicol (30 μg), erythromycin (15 μg), gentamicin (10 μg), penicillin G (1 µg), trimethoprim-sulfamethoxazole (1.25/23.75 μg), and tetracycline (30 μg) (Oxoid, Basingstoke, UK). The inducible resistance to macrolide-lincosamide-streptogramin B (MLSB) was detected by the disk diffusion method using the clindamycin (2 μg) and erythromycin (15 μg) disks positioned 15–26 mm apart [45]. Multidrug resistance (MDR) was defined as a resistance to three or more classes of antimicrobials.

Methicillin resistance was initially identified using cefoxitin (30 µg) and the detection of PBP2a protein (OXOID™ PBP2 ‘Latex Agglutination Test Kit, Basingstoke, England), and finally confirmed by the detection of the *mec*A gene according to Khairalla et al. [33]. *S. aureus* ATCC 25,923 (methicillin-susceptible) and *S. aureus* ATCC 43,300 (methicillin-resistant) were used as the reference strains.

### 4.4. Statistical Analysis

The significance of between-group differences was verified with the Pearson chi-squared test or Fisher exact test with Statistica 14 (StatSoft, Tulsa, OK, USA). A *p*-value < 0.05 was considered statistically significant.

## 5. Conclusions

This study showed that preclinical dentistry students have a similar rate of oral *S. aureus* carriage as the nasal rate, and that the oral cavity can be colonized by antimicrobial-resistant strains that do not originate from the nose. Consequently, the oral cavity seems to be an unjustly overlooked body site in screening for *S. aureus* carriage. Moreover, students should be well educated about the risk of spreading antimicrobial-resistant *S. aureus*, especially MRSA strains. 

## Figures and Tables

**Table 1 antibiotics-13-00649-t001:** Oral and nasal carriage of *S. aureus* among preclinical dentistry students.

Year of Study	Students (n)	Carriers % (n)	Oral Carriers % (n)	Nasal Carriers % (n)	Oral and Nasal Carriers % (n)
I year	67	52.2 (35)	10.4 (7)	11.9 (8)	29.9 (20)
II year	65	43.1 (28)	7.7 (5)	10.8 (7)	24.6 (16)
Total	132	47.7 (63)	9.1 (12)	11.4 (15)	27.3 (36)

**Table 2 antibiotics-13-00649-t002:** Antibiotic resistance rates of nasal and oral *S. aureus* isolates.

Antibiotics	Nasal Isolates % (n = 57)	Oral Isolates % (n = 53)	Total % (n = 110)
Cefoxitin (FOX)	1.8 (1)	3.8 (2)	2.7 (3)
Penicillin G (P)	94.7 (54)	96.2 (51)	95.5 (105)
Tetracycline (TE)	47.4 (27)	47.2 (25)	47.3 (52)
Erythromycin (E)	29.8 (17)	26.4 (14)	28.2 (31)
Clindamycin (DA)	21.1 (12)	17.0 (9)	19.1 (21)
Gentamycin (CN)	15.8 (9)	15.1 (8)	15.5 (17)
Trimethoprim-sulfamethoxazole (SXT)	3.5 (2)	1.9 (1)	2.7 (3)
Ciprofloxacin (CIP)	(0)	(0)	(0)
Chloramphenicol (C)	(0)	(0)	(0)

## Data Availability

Data are contained within the article.

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
