# Peer review of "The Oral Cavity—Another Reservoir of Antimicrobial-Resistant Staphylococcus aureus?"

_antibiotics, 2024, doi:10.3390/antibiotics13070649_

Round 1
Reviewer 1 Report
Comments and Suggestions for Authors
The article is too short to be considered a research article!
Comments on the Quality of English LanguageThe English quality is fine
Author Response
Dear Reviewer,
We would like to thank you for all of your valuable remarks.
We have addressed them all in detail below.
With kind regards,
Authors
The article is too short to be considered a research article!
Re: The manuscript has been transformed into a Brief Report.
Reviewer 2 Report
Comments and Suggestions for Authors
A well designed and timely study on an assessment of the prevalence and antimicrobial susceptibility of S. aureus in both oral and nasal cavities among preclinical dentistry students. The results show the samples from the oral reservoir are usable in anti-bacterial studies and give a new route to relevant work.
My one concern is that the English is clearly 'machine corrected. The authors need an aged native English speaker to correct the manuscript. One accomplished can be published.
Comments on the Quality of English Language
My one concern is that the English is clearly 'machine corrected. The authors need an aged native English speaker to correct the manuscript. One accomplished can be published.
Author Response
Dear Reviewer,
We would like to thank you for the overall positive evaluation of our manuscript.
Proofreading through the text for the revised manuscript has been accomplished.
With kind regards,
Authors
Reviewer 3 Report
Comments and Suggestions for Authors
A respectful greeting and congratulations to the authors of the paper, the question posed is important and useful, because it makes us think of a way to determine the presence of colonization by S. aureus that can be useful in the clinical context. in addition to this, it uses a population that deserves to be study in more detail, such as health workers and particularly dentists, some findings that catch my attention and that when contrasted with the reality of microbiology in other regions are the low found rate of resistance to methicillin and it seems to me that this merits some additional comment. I also believe that not finding resistance to quinolones deserves special consideration because it would be interesting to know what the circulating clone of said microorganism is in the geographical area, country or locality where the study is carried out, knowledge of the latter has prognostic considerations for example and I would be important to raise some hypothesis about the finding of multidrug resistance more frequently found in oral cavity isolates.
Author Response
Dear Reviewer,
We would like to thank you for the overall positive evaluation of our manuscript and all of your valuable remarks. We have addressed them all in detail below.
Resistance to fluoroquinolones in Poland is low, with 1.4% of hospital-acquired infections [1].
In order to explore circulating antibiotic-resistant S. aureus clones, we will subject isolated strains to spa typing, as stated in Discussion paragraph line 170.
With kind regards,
Authors
Reference:
- European Centre for Disease Prevention and Control (ECDC). TESSy – The European Surveillance System – Antimicrobial resistance (AMR) reporting protocol – European Antimicrobial Resistance Surveillance Network (EARS-Net) surveillance data for 2021. Stockholm: ECDC; 2022
Round 2
Reviewer 1 Report
Comments and Suggestions for Authors
The article is fine after revision